# Liposomal Bilayer as a Carrier of *Rosa canina* L. Seed Oil: Physicochemical Characterization, Stability, and Biological Potential

**DOI:** 10.3390/molecules28010276

**Published:** 2022-12-29

**Authors:** Aleksandra A. Jovanović, Danica Ćujić, Bojan Stojadinović, Natalija Čutović, Jelena Živković, Katarina Šavikin

**Affiliations:** 1Institute for the Application of Nuclear Energy INEP, Banatska 31b, Zemun, 11080 Belgrade, Serbia; 2Institute of Physics, Pregrevica 118, Zemun, 11080 Belgrade, Serbia; 3Institute for Medicinal Plants Research “Dr Josif Pančić”, Tadeuša Košćuška 1, 11000 Belgrade, Serbia

**Keywords:** liposomes, size, Raman spectroscopy, *Rosa canina* seed oil, stability

## Abstract

*Rosa canina* L. seeds are rich in bioactive components that can add value to the various formulations. The focus of the study was the development of liposomes for *R. canina* oil to protect its sensitive compounds and prolong their shelf-life. Oil-loaded liposomes were characterized via the determination of the particle size, polydispersity index (PDI), zeta potential, conductivity, mobility, density, surface tension, viscosity, and stability. Raman and FT-IR spectroscopy were employed to investigate the chemical composition of the non-treated and UV-treated samples, and the presence of different interactions. Antioxidant and antimicrobial activities were examined as well. The liposome size was 970.4 ± 37.4 nm, the PDI 0.438 ± 0.038, the zeta potential −32.9 ± 0.8 mV, the conductivity 0.068 ± 0.002 mS/cm, the mobility −2.58 ± 0.06 µmcm/Vs, the density 0.974 ± 0.004 g/cm^3^, the surface tension 17.2 ± 1.4 mN/m, and the viscosity 13.5 ± 0.2 mPa•s. The Raman and FT-IR spectra showed the presence of lipids, fatty acids, polyphenols, and carotenoids. It was approved that the oil compounds were distributed inside the phospholipid bilayer and were combined with the membrane interface of the bilayer. The UV irradiation did not cause any chemical changes. However, neither the pure oil nor the oil-loaded liposomes showed any antimicrobial potential, while the antioxidant capacity of the oil-loaded liposomes was significantly low. The sizes of the liposomes did not change significantly during 60 days of storage. Due to the proven stability of the oil-loaded liposomes, as well as the liposome’s ability to protect the sensitive oil compounds, their potential application in the pharmaceutical and cosmetic formulations could be investigated with a focus on the skin regeneration effects.

## 1. Introduction

Rosehips are pseudo-fruits from the plants of the *Rosa* genus in the Rosaceae family that possess vitamin C, carotenoids, pectic substances, polyphenols, riboflavin, sugars, lipids, plant acids, and fatty oil [1,2]. They are known to have antioxidant, antimicrobial, anti-inflammatory, anti-diabetic, and anticancer effects [1]. *Rosa canina* is the major commercial source of rosehips. Furthermore, rosehip seeds, as a waste (by-product) from the manufacture of rosehip juice or syrup, contain 4.9–17.8% of fatty oil [3]. *R. canina* seed oil contains polyunsaturated fatty acids, palmitic, stearic, oleic, arachidic, linoleic, and linolenic acids, carotenoids, phenolic acids, tocopherols, squalenes, minerals, and phytosterols, particularly β-sitosterol [4,5]. Plant oil with a high concentration of unsaturated fatty acids possesses the potential health benefits but shows a higher susceptibility to lipid oxidation, as well [6].

*R. canina* seed oil possesses antioxidant, antimicrobial, and anticancerogenic activities, shows a positive influence on dermatoses, ulcers, and other skin diseases, and represents a valuable source of phytonutrients that can improve the lipid metabolism [3,4,7], thus it is becoming popular in the cosmetic, pharmaceutical, and agro-food industries. According to the exploratory study conducted by Shabykin and Godorayhi [8], *R. canina* seed oil may be a promising topical agent for the treatment of eczema, neurodermatitis, cheilitis, and trophic ulcers of the skin. Grajzer et al. [3] have reported that the oil is a valuable source of linolenic acid, lipophilic antioxidant compounds, particularly γ- and δ-tocopherol, as well as carotenoids. The same study has shown its high radical scavenging activity and high content of other unsaturated fatty acids. The microbiological shelf life of the fillets exposed to the *R. canina* seed oil nanoemulsion was prolonged, due to the antimicrobial properties of the oil [9]. According to Butnaru et al. [7], the addition of *R. canina* seed oil in the chitosan-based bionanocomposite films, increased the antibacterial activity against *Escherichia coli*.

Since *R. canina* seed oil contains various flavonoids, phenolic acids, carotenoids, and unsaturated fatty acids that can be sensitive to oxygen, light, UV irradiation, enzymes, and pH value variations, and possess a low bioavailability, the carriers for the oil should be developed to protect the bioactive compounds and provide their higher and controlled release, as well as a more comfortable oral or dermal applications. Namely, Mudrić et al. [6] have shown that after 12 days of storage at 65 °C (corresponding to the one-month storage at 25 °C, according to Chong et al. [10]), in pure *R. canina* seed oil without additional antioxidants, the polyphenol content and radical scavenging activity significantly decreased (~50% and ~1.5%, respectively), while in the oil sample with the polyphenol extract, the polyphenol content, and the antioxidant capacity significantly rose. Additionally, after 12 days of storage at 65 °C, the content of palmitic, stearic, oleic, eicosenoic, and *cis*-11-eicosenoic acid was significantly higher in the oil sample with the polyphenol extract than in the pure oil [6]. Additionally, due to the low solubility in aqueous surroundings, easy oxidization, and poor bioavailability, the application of rosehip fatty oil in hydrophilic food, pharmaceutical, and cosmetic formulations has been limited. Thus, rosehip seed oil requires encapsulation for further application with the aim to protect its sensitive bioactive compounds.

Liposomes have been used as a carrier for delivering enzymes, polyphenols, drugs, proteins, vitamins, aromas, and antioxidant compounds [11,12,13,14]. The main advantages of liposomes over other encapsulation procedures are the stability that liposomes provide in products with a typically high water content and the ability of liposomes to encapsulate the hydrophilic, amphiphilic, and lipophilic compounds [15,16]. In addition, liposomes represent non-toxic and biodegradable delivery systems that are usually prepared from naturally occurring compounds, thus new formulations could be easily implemented [11,15]. The liposomal bilayer also provides the enhanced bioavailability of various drugs [17,18], protein/peptide-based therapeutics [17], nutraceuticals [19], and polyphenols [17,20,21]. Proliposome technologies provide the high energy input of agitation and thus smaller and uniform liposomes, compared to the other procedures for the liposomal preparation [11]. Furthermore, the proliposome method may be suitable for producing liposomes on a large scale [22]. Since the textural, rheological, and physical properties, as well as chemical characteristics of the topical and oral drug delivery systems, have a direct influence on the drug bioavailability, as well as on the manufacture of pharmaceutical or cosmetic formulations, the mentioned characteristics of the liposomes should be investigated. The Raman and Fourier transform infrared (FT-IR) spectroscopy are widely used for the investigation of the interaction of the bioactive compounds, drugs, vitamins, or hormones with a phospholipid liposomal bilayer [23,24,25,26,27,28].

The encapsulation of fatty oils in liposomal particles was the focus of several recent studies [29,30,31]. Since there is no comprehensive study regarding the encapsulation of *R. canina* seed oil into liposomes, in the present study, the oil-loaded liposomes were developed and characterized with the aim to protect the sensitive biologically active components of the oil, increase their bioavailability, and provide their controlled release, thus be potentially implemented into various pharmaceutical and cosmetic formulations. Specifically, the particle size, polydispersity index (PDI), zeta potential, conductivity, mobility, density, surface tension, viscosity, the Raman and FT-IR spectra, and the 60-day storage stability of the liposomes (non-treated and UV-irradiated) were examined. Additionally, the determination of their antioxidant and antimicrobial activities was also performed. Therefore, the presented study can provide evidence of the physicochemical properties and the biological activity of the *R. canina* seed oil-loaded liposomes that can add value and improve the quality of the pharmaceutical and cosmetic formulations.

## 2. Results and Discussion

### 2.1. Size, PDI, Zeta Potential, Conductivity, and Mobility of the Non-Treated and UV-Irradiated Liposomes

The particle size, PDI, zeta potential, conductivity, and mobility of the empty and *R. canina* seed oil-loaded liposomes (non-treated and UV-irradiated) were determined using photon correlation spectroscopy (PCS). The results obtained immediately after the liposomal preparation and after the UV irradiation are shown in Table 1.

As can be seen in Table 1, the size of the oil-loaded liposomes measured immediately after the liposomal preparation was 970.4 ± 37.4 nm, while the PDI was 0.460 ± 0.010. According to the literature, the procedures that provide a high energy input of the agitation, which was the case of the proliposome technique used in the present study, form lower-sized particles, compared to the other methods, such as the thin film procedure [11,15]. Isailović et al. [11] have reported that the presence of a small amount of ethanol which is used in the proliposome method affects the liposome size, in terms of reducing the parameter. Namely, ethanol causes the modification of the system net charge, causing a steric stabilization. The liposomes with oil had a significantly smaller diameter than their empty parallel (2145.7 ± 43.7 nm, Table 1). Namely, according to Fathi-Azarbayjani et al. [32], the encapsulation of the oil compounds within the liposomal bilayer reduces the number of phospholipids that are incorporated into the membrane and therefore causes the formation of smaller vesicles. The obtained PDI value of the plain and oil-loaded liposomes (~0.460, Table 1), as a measure of the particle size distribution, indicates the existence of a moderately disperse distribution [33]. Zhao et al. [34] have reported that higher concentrations of lipids, particularly sterols, led to the higher PDI values showing an increased heterogeneity. Considering the high concentration of phospholipids in the liposomal samples (0.2 g per mL), as well as the presence of phytosterols from rosehip oil in oil-loaded liposomes, it can be the reason for the higher PDI values, i.e., a moderately disperse size distribution. UV irradiation did not cause statistically significant changes in the vesicle size and uniformity of the system (Table 1).

The zeta potential determined immediately after the preparation of the oil-loaded liposomes was −32.9 ± 0.8 mV (Table 1). The zeta potential can significantly impact the physical stability of the vesicles in suspension by determining the electrostatic repulsion between them [11]. Thus, the negative and high value of the zeta potential (as a measure of liposomes stability) determined in our case, accounts for a good electrostatic stabilization of the system, preventing the aggregation and fusion of the particles [11,15]. Additionally, the surface charge is a key determinant of the nano-particle and encapsulated content behavior and the elimination in vivo. The anionic nano-particles interact strongly with the reticuloendothelial system cells scavenging the endothelial cells and the blood resident macrophages, whereas the cationic liposomes are rapidly removed from circulation by a combination of non-specific cellular interactions (adsorption to the anionic surface of the blood vessel walls), and clearance by the specialized cells of the reticuloendothelial system [35]. The rosehip oil-loaded liposomes showed a higher zeta potential (absolute value), compared to the plain liposomal bilayer (−22.7 ± 0.5 mV, Table 1). Namely, several research studies have reported that the sterol incorporation increased the spacing between the phospholipid heads and caused the hydrophobic stabilization of the membrane of the liposomes [36,37,38]. Therefore, the presence of phytosterols from rosehip oil can change the order of the phospholipids and the thickness of the liposomal membrane regardless of the nature of the functional groups in phospholipids, and these groups can participate in creating hydrogen bonds with phytosterols, as well as in changing the zeta potential of the liposomes [36,39]. In addition to the particle size, the PDI, and the zeta potential, the conductivity of the prepared liposomes was determined as the fourth physical property and amounted to 0.068 ± 0.005 mS/cm for the oil-loaded liposomes (Table 1). The electrical conductivity represents a measure of how easily an electrical current can pass through water or any solution, and it can be used as an indicator of the total dissolved solids [40,41]. In the case of the liposomes, the conductivity is influenced by the exposed charge of the phospholipids and correlates to a volume of the liposome entrapment [41]. The conductivity measurements can also provide evidence of polyions that can induce the liposome aggregation. The additional reasons for the monitoring of the conductivity of the liposomal suspension lay in the fact that the movement of the conductive ions in highly conductive samples can lead to the electro depolarization, degradation, and thus inconsistent zeta potential values, while changes in the conductivity values (the increase in values) may indicate a leakage of the encapsulated contents into the surrounding water medium [40]. Namely, Lidgate et al. [41] have reported that in the case of a higher phospholipid concentration (which was the case of rosehip oil-loaded liposomes, 0.2 g/mL), the ions are inside the liposomal particles, their mobility is reduced, and their contribution to the conductivity is no longer apparent, thus the value of the conductivity is pretty low. However, the conductivity of the plain liposomes was even significantly lower (0.020 ± 0.002 mS/cm), in comparison to the oil-loaded liposomes. It can be explained by the fact that the small liposomes most notably affected the conductivity, due to a greater surface area which exposes a greater percentage of the phospholipid head groups [41]. Thus, the measured conductivity of the *R. canina* oil-loaded liposomes was greater (smaller particles than of plain liposomes). The mobility of the liposomes, as a function of size, surface charge, and membrane composition, was also examined (−2.58 ± 0.06 µmcm/Vs, Table 1). In our previous study [15], the liposomes with sterols showed a higher fluidity, thus a better mobility. Given that the *R. canina* seed oil possessed phytosterols, mainly β-sitosterol—82.1% [4], it can be the explanation for the higher mobility of the oil-loaded liposomes, compared to the plain liposomes (−1.79 ± 0.03 µmcm/Vs, Table 1). However, the UV irradiation has caused a significant influence on the zeta potential, conductivity, and mobility of the oil-loaded liposomes. Namely, a drop in the all mentioned values was noticed (Table 1). According to the literature, the UV irradiation of the liposomes resulted in the comprehensive surface charge change or even reversal from the negative to the positive [35]. However, the zeta potential changing or switching do not always induce any large-scale reorganization of the liposomal bilayer, disruption of the liposome integrity, as well as leakage of the encapsulated bioactive compounds. Considering that there was no increase in the conductivity of the oil-loaded liposomes after the UV irradiation, it can be concluded that there was no leakage of the encapsulated compounds from the carrier. Namely, the increase in the conductivity of the liposomes during storage is usually related to the leakage of the encapsulated active compounds. The conductivity of the inner medium of the liposomes can be modified by the encapsulated compounds as well [42]. Since 30-min UV irradiation caused the water evaporation from the liposomal suspension, the decrease in the conductivity of the UV-irradiated samples should be explained by higher lipid concentrations which lead to a higher capture volume, the effective removal of ions from the liposome suspension, and consequently, to the reduction in the conductivity value [32,41]. A drop in the mobility can be explained by the degradation of β-sitosterol from the oil because the UV irradiation can induce the phytosterol peroxidation [43].

### 2.2. The Storage Stability of the Liposomes

With the aim to investigate the storage stability of the *R. canina* seed oil-loaded liposomes (non-treated and UV-irradiated), the particle size, the PDI, the zeta potential, the conductivity, and the mobility were determined for 60 days and the results are presented as Figure 1a–e. The storage stability of the empty liposomes (non-treated and UV-irradiated) via analyzing the particle size, PDI, zeta potential, conductivity, and mobility were also examined for 60 days and the results are presented in Appendix A.

The size of the liposomal population with the rosehip oil did not change significantly during 14 days of storage at 4 °C, but after the 21st day, the vesicle size rose from 952.8 ± 20.1 nm to 1157.7 ± 25.8 nm and amounted to 1179.2 ± 21.4 nm at the 60th day, whereas the PDI continuously increased from the 14th day (Figure 1b). As it is mentioned in Section 3.1., a negative zeta potential of the oil-loaded liposomes provides a good electrostatic stabilization of the liposomal suspension, providing the inhibition of the particle aggregation and fusion, which is confirmed by the stability test (the size did not change for more than 11.5% during the 60-day storage at 4 °C). In the UV-irradiated parallel, the same trend is observed.

As can be seen in Appendix A, on the 1st day of the storage stability study, the non-treated vesicles of the plain liposomes had the same diameter as the UV-irradiated vesicles. However, after 60 days of storage at 4 °C incubation, the vesicle diameters changed only for the UV-irradiated sample. Namely, on the 60th day, the diameter for UV-irradiated liposomes was 2650.5 ± 27.1 nm and for the non-treated 2115.4 ± 45.5 nm. The size of the UV-irradiated liposomes changed by 54.6% during the 60-days storage at 4 °C. The obtained results are in agreement with the literature data [44]. In both plain liposomes, the PDI did not change during the storage stability study (Appendix A).

Moreover, the zeta potential varied in both the oil-loaded liposomes, but the trend was different in the non-treated and the UV-irradiated samples (Figure 1c). The zeta potential (absolute value) in the non-treated liposomes decreased from −32.9 ± 0.8 mV to −22.2 ± 0.9 mV, while the zeta potential of the UV-irradiated liposomes decreased to −18.8 ± 0.3 mV on the 14th day, and after that it increased up to 21.8 ± 0.9 mV on the 60th day. The zeta potential of the non-treated plain liposomes did not change significantly during the 60-day storage study, while in the case of the UV-irradiated parallel, the zeta potential increased from −17.6 ± 0.3 mV to −23.8 ± 0.5 mV (Appendix A). According to the results of the liposomes’ mobility (Figure 1d), it can be concluded that there were no significant changes in the mentioned parameters during the 21 days in the non-treated oil-loaded liposomes, but after that, the mobility decreased up to −1.70 ± 0.06 µmcm/Vs. However, the mobility of the UV-irradiated parallel did not change during the 60-day storage. Moreover, in the UV-irradiated plain liposomes, an increase in the mobility of the liposomes after 14 days (from −1.22 ± 0.02 µmcm/Vs to −1.71 ± 0.03 µmcm/Vs) is observed, while in the non-treated plain liposomes, there were no changes in the mobility during the 60 days. Therefore, the trend of the zeta potential changes, is the same as the trend of the mobility changes in the case of the unloaded liposomes.

According to Lidgate et al. [41], the measurement of conductivity of the stored liposomal vesicles can provide important information related to their size and integrity, i.e., the potential leaking of the encapsulated substances. Namely, the changes in conductivity can indicate whether the liposomes have fused, i.e., the decrease in conductivity, or leaked, i.e., an increase in conductivity. The conductivity of both non-treated and UV-irradiated oil-loaded liposomes (Figure 1e) significantly decreased during the 60-day stability study (from 0.068 ± 0.002 mS/cm to 0.007 ± 0.003 mS/cm for the non-treated and from 0.032 ± 0.001 mS/cm to 0.009 ± 0.001 mS/cm for the UV-irradiated). The same trend is observed in the case of the plain liposomes (Appendix A). As can be noticed, the conductivity of the oil-loaded liposomes decreased on the 14th day, after that the values did not change. A similar trend was observed in the PDI values of the mentioned liposomes (the increase in the PDI on the 14th day, after that the values were the same). Therefore, the decrease in the conductivity can be related to the fusion of some parts of the liposomal population, which influenced the size distribution, but not particle size represented as the mean value (Figure 1a,b).

### 2.3. Density, Surface Tension, and Viscosity of the Liposomes

The physical properties, including the density, surface tension, and viscosity were determined in the pure *R. canina* seed oil, the empty, and the oil-loaded liposomes (non-treated and UV-irradiated). Since the mentioned parameters have a key role in the manufacture of pharmaceutical and cosmetic products, they should be investigated, in order to improve the product quality.

As can be seen in Table 2, the density of the oil-loaded liposomes was 0.974 ± 0.003 g/cm^3^ for the non-treated and 0.959 ± 0.001 g/cm^3^ for the UV-irradiated, whereas the plain liposomes have shown a significantly higher density, 0.995 ± 0.003 g/cm^3^ and 0.974 ± 0.004 g/cm^3^ for the non-treated and the UV-irradiated samples. The surface tension of the oil-loaded liposomes was 20.3 ± 1.0 mN/m for the non-treated and 23.1 ± 0.2 mN/m for the UV-irradiated, while the unloaded liposomes had a significantly different surface tension, 19.7 ± 0.9 mN/m for the non-treated and 21.4 ± 0.5 mN/m for the UV-irradiated. The viscosity of the oil-loaded liposomes was 13.5 ± 0.2 mPa•s for the non-treated and 20.2 ± 0.1 mPa•s for the UV-irradiated, while the viscosity of the empty liposomes was 13.9 ± 0.2 mPa•s for the non-treated and 20.7 ± 0.4 mPa•s for the UV-irradiated. Further, the density and surface tension of the pure oil were 0.914 ± 0.001 g/cm^3^ and 30.0 ± 0.8 mN/m for the non-treated and 0.915 ± 0.002 g/cm^3^ and 28.0 ± 0.9 mN/m for the UV-irradiated, while viscosity was 50.9 ± 0.3 mPa•s for the non-treated and 45.7 ± 0.1 mPa•s for the UV-irradiated. In the case of the pure oil, the UV irradiation has caused a decrease in the surface tension and viscosity values. Moreover, it was the opposite in the oil-loaded liposomes, where the UV-irradiated samples possessed a higher surface tension and viscosity, probably due to water evaporation during 30 min of UV irradiation. The values of the density and surface tension did not change after the 60th day in the non-treated and UV-irradiated oil-loaded liposomes (0.969 ± 0.003 and 0.962 ± 0.003 g/cm^3^ and 20.1 ± 1.5 and 23.4 ± 1.5 mN/m, respectively), while the viscosity significantly decreased (6.7 ± 0.4 and 11.4 ± 0.5 mPa•s), probably due to the hydrolytic reactions characteristic of an aqueous medium.

### 2.4. Raman Spectra

The Raman spectroscopy was applied as the fast and nondestructive analytical technique for the chemical evaluation and the examination of the presence of different interactions between Phospholipon (Ph, a commercial phospholipid mixture) and the *R. canina* seed oil, as well as the changes between the plain and oil-loaded liposomes, and their UV-treated parallels.

The Raman features of the Ph and plain liposomes spectra (Figure 2a and Appendix A, respectively) correspond to the C-N symmetric stretching of choline (phospholipid head-group) at ~710 cm^−1^, C–C=O stretching at 959 cm^−1^, the skeletal stretching of the C–C vibrations at ~1025 cm^−1^, the symmetric stretching of PO_2_^−^ at 1080 cm^−1^, the asymmetric stretching region of the PO_2_^−^ groups between 1170–1200 cm^−1^, *cis*=C-H stretching vibration in the oleoyl chain at 1275 cm^−1^, the methylene deformation band (*δ*CH_2_) of the fatty acid chains at 1450 cm^−1^, C=C stretching vibration in the oleoyl chain at 1665 cm^−1^, carbonyl group (C=O) of the ester bond between glycerol and fatty acids at 1765 cm^−1^, the symmetric and asymmetric stretching modes of the C–H bonds in CH_2_ and CH_3_ groups in the alkyl chains at 2850–2950 cm^−1^, the CH stretching of the N-CH_3_ groups at 3015 cm^−1^, and the mode associated with bound water at 3325 cm^−1^ [23,24,28,45,46]. Additionally, the Raman mode at ~1525 cm^−1^ can be related to the N-O stretching (probably from the impurity) that occurs at 1500–1550 cm^−1^ or the H-O-H bending vibration in the phospholipids that occurs between 1500 and 1700 cm^−1^ [45]. The difference between Ph and the UV-irradiated Ph is in a slight moving from 1275 cm^−1^ to 1260 cm^−1^ of the band ascribed to the *cis* =C-H stretching vibration in the oleoyl chain, probably due to the sensitivity of a double bond to the oxidation caused by the UV irradiation. The spectra of the plain liposomes (non-treated and UV-irradiated), presented in Appendix A, contain all peaks as the spectra of pure Ph, thus it can be concluded that no chemical changes occur in the phospholipids during the liposome formulation, which is in agreement with the literature data [23]. However, there were differences in the intensity of the peaks between the spectrum of pure Ph and the plain liposomes, possibly due to the formation of hydrogen bonds. Arsov and Quaroni [47] and Chen and Tripp [48] have reported that the relative intensity of the C=O bands is the key parameter to monitor the changes in the relative free and hydrogen bonded populations of the carbonyl groups of the lipid.

The fingerprint region of the Raman spectra of the rosehip oil includes essential bands which correlate with the most important parts of the fatty acids’ molecular structure and the region well known to characterize the unsaturation level of the fatty acid chain [49,50]. According to the literature, the dominant compounds in the *R. canina* seed oil are unsaturated acids, including linoleic, α-linolenic, and oleic acids, while linoleic acid possesses the highest percentage [4,51,52]. Moreover, saturated fatty acids, such as palmitic and stearic acids, are presented in a lower percentage [53]. Linoleic, α-linolenic, and oleic acids mainly differ in the position of the double bond, and consequently, there are two or three broad C=C bonds with higher wavenumbers in their Raman spectra that are highly similar [54,55]. As can be seen in Figure 2, a group of overlapping bands in the region between 800 and 1100 cm^−1^, peaking at 1010 cm^−1^, may be attributed to the C–C stretching of the (CH_2_)_n_ group [56]. A strong Raman phonon in the carbonyl region of the spectrum at 1160 cm^−1^ can be assigned to the C–C stretching of carotenoids [23,56]. The mode at 1275 cm^−1^ is related to the presence of the esterified unsaturated fatty acids (*cis* isomers) and can be assigned to the bending of =C-H [53]. The band at 1450 cm^−1^ can be assigned to the C–H scissoring of CH_2_, while the peak at 1520 cm^−1^ can be assigned to the C=C stretching of carotenoids [56]. A broad small peak at ~1600 cm^−1^ corresponds to the polyphenolic compounds [53], which is in agreement with the study of Grajzer et al. [3] who identified and determined, apart from a relatively high level of carotenoids in rosehip seed oil, phenolic acids, particularly *p*-coumaric acid methyl ester, vanillin, and vanillic acid. Since rosehip oil possesses an appreciable amount of lipophilic antioxidants, such as tocopherol [3], the phonon arising from the aromatic part (chromanol ring) of tocopherol occurred at 1615 cm^−1^ [57]. The phonon frequency at 1650 cm^−1^ can be assigned to the *cis* stretching vibration of the C=C unsaturated lipids (*cis* RHC=CHR) [53,56]. The relative intensity of all unsaturation regions of the unsaturated fatty acids (*cis* isomers) is in accordance with the degree of saturation of the fatty acid in the lipid, especially in the case of the mode at 1650 cm^−1^ [53]. A broad band in the oil spectrum ranging at ~2700 cm^−1^, originates from CH_2_, and the CH_3_ symmetric stretching vibration [58]. The UV irradiation did not cause changes that can be visible in the Raman spectra of the pure *R. canina* oil. The obtained spectra are in agreement with the literature data and contain the common characteristic vibrational modes of almost all vegetable oils [56].

The Raman spectrum of the rosehip seed oil-loaded liposomes showed characteristic bands of both phospholipids and oil (Figure 2b). Namely, the mode at 959 cm^−1^ corresponds to the phospholipid C–C=O stretching. In a region of 1010–1025 cm^−1^, the C–C stretching of the (CH_2_)_n_ group presents in Ph and oil spectra, while the mode 1150 cm^−1^ can be assigned to the C–C stretching of carotenoids, which exclusively originates from oil. However, the stretching region of the PO_2_^−^ groups is from 1170 to 1200 cm^−1^. Namely, in the oil-loaded liposomes spectrum, the stretching region of the PO_2_^−^ groups of phospholipids was found at 1135–1195 cm^−1^ (occurred in the spectrum of raw phospholipids as well, Figure 2a) and it seems to be insensitive to the presence of either bilayer modifiers, such as rosehip oil. Namely, phytosterols from plant oil, such as β-sitosterol and cycloartenol presented in rosehip oil [3], are incorporated within the bilayer membrane, its hydrophilic 3β-hydroxyl head group is placed in the vicinity of the phospholipid ester carbonyl groups, and its hydrophobic steroid ring is oriented parallel to the acyl chains of the phospholipids [24]. The mode at ~1275 cm^−1^ relates to the *cis* =C-H stretching vibration in the oleoyl chain and methylene deformation band (*δ*CH_2_) of the fatty acid chains at 1450 cm^−1^ presented in both Ph and oil. The Raman phonon at 1525 cm^−1^ can be related to the C=C stretching of carotenoids from oil, whereas the C=C stretching vibration in the oleoyl chain at 1650 cm^−1^ is from unsaturated fats presented in Ph and oil. The symmetric and asymmetric stretching modes of the C–H bonds in the CH_2_ and CH_3_ groups in the alkyl chains are in a range of 2700 to 3000 cm^−1^ which occurred in the Ph and oil spectra as well. The obtained results are expected since rosehip oil had a very strong signal during the Raman spectra recording. Additionally, in contrast to the aqueous extracts which compounds are water soluble and completely enveloped by the phospholipid layers [23], the liposoluble components of the oil cannot be affected by the effect of “shielding” [59]. Namely, the lipophilic compounds are located within the liposomal bilayer that contains phospholipid tails, whereas the hydrophilic compounds can be located in the water vesicles [15]. According to Pohle et al. [60], the stretching region of the PO_2_^−^ groups at 1080 cm^−1^ and ~1200 cm^−1^ is marked as being quite sensitive to the structural changes of the phosphatidylcholine micelles, particularly in the hydrated state. In the case of the oil-loaded liposomes, a mode at 1080 cm^−1^ disappeared, whereas the peak at ~1200 cm^−1^ became more pronounced and shifted to the higher wavenumber. In the study by Frías et al. [26], a polyphenol from bearberry, encapsulated into phosphatidylcholine liposomes, influenced both of the PO_2_^−^ stretching phonons’ frequency, depending on its location within the liposomal vesicles. Namely, the mode associated with the symmetric PO_2_^−^ stretching at 1080 cm^−1^ was affected when the polyphenol was outside, whereas the asymmetric stretching phonon at ~1200 cm^−1^ was influenced if the polyphenol is both inside and outside. Additionally, the encapsulation of the polyphenols of ground ivy extract in the liposomal bilayer also caused a slight shift of the PO_2_^−^ asymmetric stretching band from 1238 cm^−1^ (empty liposomes) to 1243 cm^−1^ (extract-loaded liposomes) [23]. In the case of rosehip seed oil-loaded liposomes, changes have occurred in both the PO_2_^−^ stretching regions indicating that the oil compounds are located both in the phospholipid bilayer and on the surface of the membrane. An additional fact that some of the rosehip oil components can be distributed at the liposomal surface is the diminishing of the intensity of the phonon related to the bound water at 3325 cm^−1^. Namely, the components that interact via the hydrogen bonds can replace molecules of water adsorbed on the membrane surface. The facts that come from the Raman analysis suggest that the oil components are not only entrapped within the liposomal bilayer but also on the membrane surface, having a significant influence on the physicochemical properties of the liposomes. As in the case of the plain liposomes and pure oil, the UV irradiation did not cause changes that can be visible in the Raman spectra of the oil-loaded liposomes.

### 2.5. FT-IR Spectra

The FT-IR spectroscopy was also applied in order to analyze and investigate the presence of different interactions between the phospholipids from liposomes and *R. canina* seed oil, as well as the potential influence of the UV irradiation on the liposomes and oil compounds. The FT-IR spectra of the phospholipid mixture, rosehip oil, plain and oil-loaded liposomes, and their UV-irradiated parallels are presented in Figure 3.

Regarding the FT-IR spectra of the pure Ph (Figure 3a), a group peak at 623 cm^−1^ assigned to O–CO–C of phospholipids, C-N symmetric stretching of choline at ~719.6 cm^−1^ is observed, and the quaternary ammonium group of choline moiety gives rise to two phonons, at 923.1 and 969 cm^−1^, which is related to the symmetric and asymmetric stretching of the C–N bond, respectively [23,24]. There is a band at 1086.4 cm^−1^ that corresponds to the symmetric PO_2_^−^ stretching that is partially overlapped with the band at 1062.9 cm^−1^ that represents the C-O-P-O-C stretching, the mode at 1174.1 and 1233.4 cm^−1^ are specific for the symmetric and asymmetric stretching of the PO_2_^−^ groups, the mode at 1377.4 cm^−1^ is accompanied with the methylene group from the lipids, the mode centered at 1465.8 cm^−1^ assigns to the scissoring vibrations of the CH_2_ groups ascribed to the fatty acid chains, the mode at 1651.6 cm^−1^ is associated with bound water, the mode at 1735.4 cm^−1^ represents the stretching vibrations of the ester carbonyl groups (C=O), the mode at 2923.1 cm^−1^ is specific for the asymmetric stretching vibration in the CH_3_ groups, the mode at 2853.1 and 3009.8 cm^−1^ represent the symmetric and asymmetric C-H stretching, and a broad mode at 3382.9 cm^−1^ corresponds to the O-H stretching [24,61,62]. Since the FT-IR spectra of the empty liposomes (Figure 3b) showed all characteristic bands of phospholipids, it can be concluded that no chemical reaction occurred during the liposomal preparation that was also approved in the Raman analysis.

The FT-IR spectra of the *R. canina* oil (Figure 3a) show the mode at 721.8 cm^−1^ that can be assigned to the carbon skeleton vibration, and the mode at 1098.3 cm^−1^ that is characteristic for the (C–C) stretching of the (CH_2_)_n_ group [56,63]. According to Qiu et al. [56], a strong peak at 1160.5 cm^−1^ is related to the C–C stretching of carotenoids. The band at 1242.8 cm^−1^ is assigned to the in-plane deformation vibration of the =CH groups from the unconjugated cis double bonds, the phonon observed at 1377.5 cm^−1^ was associated with the deformation vibration in the phase of a methylene group from the lipids, the band at 1459.9 cm^−1^ is related to the *δ*(C–H) deformation, the phonon at 1743.2 cm^−1^ is characteristic for the *ν*(C–H) ester carbonyl, which is a chemical group specific to vegetable oils with a high content in saturated fatty acids and short hydrocarbonated chains, the mode at 2853.5 and 2923.1 cm^−1^ are assigned to the symmetric and asymmetric vibrations *ν*(C–H) of the CH_2_ and CH_3_ aliphatic groups from the alkyl rest of the triglycerides, and the mode at 3009.3 cm^−1^ is related to the =CH vibration from the lipids (methyl-oleate group) [7,63].

In the FT-IR spectra of the oil-loaded liposomes (Figure 3b), there is one mode specific only for the *R. canina* oil, at 1163.2 cm^−1^ (related to carotenoids). The presence of a low-intensity band corresponds to carotenoids, clearly originating from oil, indicating that they can be “entrapped” on the liposomal surface. Moreover, an absence of the mode that exists in the FT-IR spectra of pure oil is observed, such as the bands at 1098.3 cm^−1^ related to the (C–C) stretching of the (CH_2_)_n_ group and 1242.8 cm^−1^ assigned to the in-plane deformation vibration of the =CH groups from the unconjugated cis double bonds. Probably, the components with these functional groups are located within the liposomal particles, thus they cannot be visible in the spectra. Namely, according to the literature, when the peaks that belong to the active compounds are invisible, it means they are completely covered by the carrier, i.e., the indication of the efficient wrapping of the oil compounds during the liposome entrapment, in our case [64,65]. In the FT-IR spectra of the oil-loaded liposomes, there are also observed modes that belong to the liposome phospholipids, including the modes at 924.7, 968.6, 1089.7, and 1058.8 cm^−1^. However, the mode at 623 cm^−1^, assigned to O–CO–C of the phospholipids, 1174.1 and 1233.4 cm^−1^, related to the symmetric and asymmetric stretching of the PO_2_^−^ groups, and 3382.9 cm^−1^, corresponding to the O-H stretching, existed in the FT-IR spectra of the plain liposomes, but were not observed in the spectra of the oil-loaded liposomes. According to the literature, when the encapsulated compounds are distributed inside and outside of the liposomes, it affects the asymmetric stretching phonon at 1233 cm^−1^ [23]. In the case of the *R. canina* oil-loaded liposomes, the mentioned band is moved from 1233.4 to 1238.5 cm^−1^, indicating that the oil components are located within the liposoluble phospholipid tails and on the liposome surface. Additionally, when the encapsulated components are only on the liposomal surface, it has an influence on the band associated with the symmetric PO_2_^−^ stretching (1087 cm^−1^) [26]. Thus, further evidence that the *R. canina* oil compounds are both inside and outside of the liposomes is in the fact that the mentioned band is the same in the plain and oil-loaded liposomes. Some of the oil components that are incorporated at the membrane surface, caused a significant diminishing in the intensity of the mode assigned with bound water (at ~1650 and ~3380 cm^−1^), due to their interactions through the hydrogen bonds which replace the molecules of water adsorbed on the surface of the liposomal bilayer [23]. The obtained results and conclusions are in agreement with the literature data, where the FT-IR analysis showed that the flaxseed oil was encapsulated in the interior of the liposomes and combined with the membrane interface of the bilayer as well [30]. In addition, the same study reported that flaxseed oil loaded-liposomes were spherical with a smooth surface (transmission electron microscopy) due to the flaxseed oil which can fill the gaps created by the imperfect phospholipid arrangement and thus improve the intactness of the liposomal bilayer. Namely, since the mentioned oil compounds were combined with the membrane interface, it can explain the phenomenon that the image of the liposomes with oil had a smooth surface, while the plain liposomes had a rough surface [30]. The phonon at 1735.4 cm^−1^ (in the plain liposomes) associated with *ν*(C–H) ester carbonyl was moved to 1742.1 cm^−1^ (in the oil-loaded liposomes). The changes in the mentioned peak occurred due to the formation of a hydrogen bond between the –OH groups present in the oil components and the carbonyl groups of the phospholipids [24]. The mode that is common for both the oil and phospholipids are at ~720 cm^−1^, related to the C-N symmetric stretching and the carbon skeleton vibration, ~1377.5 cm^−1^ corresponds to the methylene group from the lipids, ~1465 cm^−1^ assigns to the scissoring vibrations of the CH_2_ groups, i.e., the *δ*(C–H) deformation ascribed to the fatty acid chains, 2853.3 and 3009.2 cm^−1^, specific for the symmetric and asymmetric C-H stretching in the lipids, and 2923 cm^−1^, related to the asymmetric stretching vibration in the CH_3_ groups.

Additionally, in all examined samples, the UV irradiation did not cause chemical changes that can be visible in the FT-IR spectra (Figure 3).

### 2.6. Antioxidant Potential of the Liposomes

The antioxidant potential of the *R. canina* seed oil, the empty and oil-loaded liposomes (non-treated and UV-irradiated), was determined using two antioxidant assays, the ABTS and DPPH methods.

As can be seen from Figure 4, the ABTS radical scavenging capacity was 0.200 ± 0.001 µmol/mL and 0.215 ± 0.008 µmol/mL, while the IC_50_ value was 0.269 ± 0.004 mg/mL and 0.215 ± 0.006 mg/mL for the non-treated and the UV-irradiated oil-loaded liposomes, respectively. The ABTS and DPPH antioxidant activities of pure oil under the same conditions amounted to 0.215 ± 0.010 µmol/mL and 0.150 ± 0.003 mg/mL, respectively. Thus, it can be concluded that the liposome surroundings did not influence the antioxidant activity of the *R. canina* oil components.

Statistically the significantly higher ABTS radical scavenging potential and the lower IC_50_ (higher DPPH radical scavenging activity) for the UV-irradiated sample with rosehip oil can be explained by the fact that the UV irradiation can induce the polymerization of the polyphenol compounds (also present in *R. canina* oil) that can have a higher antioxidant potential, in comparison to the single polyphenols [66].

As was expected, the oil-loaded liposomes have shown a statistically significant antioxidant capacity, compared to the empty liposomes, due to the presence of γ-tocopherol, δ-tocopherol, carotenoids, catechins, and phenolic acids at a relatively high level, as well as their synergistic effects that induce a significant antioxidant potential [1,3].

### 2.7. Antimicrobial Potential of the Liposomes

The antimicrobial effect of rosehip oil and oil-loaded liposomes against *Pseudomonas aeruginosa*, *Klebsiella* spp., *Proteus* spp., *Staphylococcus aureus*, and *Candida albicans,* isolated from wound swabs was investigated. However, neither the *R. canina* seed oil nor the oil-loaded liposomes showed any antimicrobial activity against all investigated microorganisms. The obtained results are in agreement with the literature data where the antibacterial effect of the *R. canina* seed oil prepared using Soxhlet extraction and *n*-hexane was not demonstrated against *P. aeruginosa*, *Staphylococcus* spp., *Lactobacillus plantarum*, *Proteus mirabilis*, *Bacillus cereus*, and *Escherichia coli*, while the methanol sample showed activity against *E. coli* [67]. According to the literature, the amount of the extracted biologically active substances of rosehip seeds strongly depends on the extraction procedures (Soxhlet, ultrasound-, microwave, sub-, and supercritical fluid extractions) and used conditions (temperature, extraction medium, pressure, time, etc.) [1,52,68]. Namely, Szentmihályi et al. [68] have reported that a higher amount of bioactive compounds was obtained using the supercritical fluid extraction and carbon dioxide/propane, in comparison to Soxhlet, ultrasound-, or microwave-extractions and *n*-hexane. Additionally, a large content of active antimicrobial compounds in the *Rosa* species is probably mainly distributed in pulp, not in seeds. Apart from that, the variation in the oil’s chemical composition was also influenced by other environmental factors, including genetic, climatic, ecologic, and the soil conditions for plant growth [1,52].

## 3. Materials and Methods

### 3.1. Plant Material, Reagents, and Standards

*R. canina* seeds were purchased from the Institute for Medicinal Plants Research “Dr Josif Pančić”, Serbia. The following reagents were used: ethanol (Fisher Scientific, Leicestershire, UK) and Phospholipon 90 G (unsaturated diacyl-phosphatidylcholine) (Lipoid GmbH, Ludwigshafen, Germany), 2,2′-azino-bis(3-ethylbenzothiazoline-6-sulphonic acid)—ABTS, *n*-hexane, Trolox—6-hydroxy-2,5,7,8-tetramethylchroman-2-carboxylic acid, 2,2-diphenyl-1-picrylhydrazyl—DPPH (Sigma Aldrich, St. Louis, MO, USA), and potassium persulfate (Centrohem, Belgrade, Serbia).

### 3.2. Preparation of the Rosa Canina Seed Oil

The oil was extracted from grounded seeds using Soxhlet extraction with n-hexane for 6 h at 60 °C. Following the evaporation of the solvent using a IKA RV 05 rotary vacuum evaporator (Staufen, Germany), the oil was stored in the dark at −25 °C.

The chemical characterization of the obtained rosehip seed oil was previously published and the content of the total polyphenols amounted to 0.13 mg of gallic acid equivalents/g, whereas the content of the unsaturated fatty acids was ~90%. The main fatty acid was linoleic (55%), followed by oleic (17.1%), α-linolenic (18.2%), palmitic (4.2%), stearic, (3.0%), eicosanoic (1.28%), *cis*-11-eicosanoic (0.36%), behenic (0.26%), and *cis*-11,14-eicosadienoic acids (0.14%) [6].

### 3.3. Preparation of the Liposomal Particles and Lyophilization

The *R. canina* seed oil-loaded liposomes were prepared using the proliposome method [11]. The liposomes were prepared using a mixture of phospholipids (Phospholipon). The phospholipids (2 g), ethanol (2.8 mL), and *R. canina* seed oil (1.2 mL) were stirred at 50 °C. Once cooled to room temperature, the aqueous phase (10 mL) was added in small portions; the emulsion was stirred at 800 rpm for 1 h. The plain liposomes (as a control) were also prepared. In the preliminary screening, the amount of oil in the liposomal preparation was varied (0.5–1.5 mL) and after centrifugation at 17,500 rpm and 4 °C for 45 min in a Thermo Scientific Sorval WX Ultra series ultracentrifuge (Thermo Scientific, Waltham, MA, USA), in the case of the samples prepared using 0.5–1.2 mL of oil, there were no oil drops on the surface of the supernatant. Therefore, 1.2 mL of the oil was selected for the preparation of the future liposomes that were further examined. For the Raman and FT-IR analyses, the lyophilized samples (empty and oil-loaded liposomes before and after the UV irradiation) were prepared as well. Following the centrifugation, the liposomes were frozen in the freezer, LAB11/EL19LT (Elcold, Hobro, Denmark), at −80 °C for 1 h and lyophilized in Beta 2–8 LD plus lyophilizator (Christ, Memmingen, Germany) at −75 °C and 0.011 mbar for 24 h.

### 3.4. Size, PDI, Zeta Potential, Conductivity, and Mobility Analyses

The particle size, PDI, zeta potential, conductivity, and mobility of the *R. canina* seed oil-loaded liposomes (immediately after the liposome preparation, after the UV irradiation, and during the 60-day study) were determined using PCS in Zetasizer Nano Series, Nano ZS (Malvern Instruments Ltd., Malvern, UK). Each sample was diluted 200 times and measured three times at 25 °C.

### 3.5. Storage Stability Study

The measurements of the particle size, PDI, zeta potential, conductivity, and mobility of the obtained liposomes were repeated on the 1st, 7th, 14th, 21st, 28th, and 60th day after the preparation of the liposomes. During the 60-day stability study, the liposomal system was stored in the refrigerator at 4 °C.

### 3.6. UV Stability Study

Since the UV irradiation is widely applied commercially for sterilization in food, pharmaceutical, and cosmetic industries, the plain and oil-loaded liposomes (3 mL in a thin layer) were exposed to UV-C irradiation (253.7 nm) in a Petri dish for 30 min [69]; subsequently, the size, PDI, zeta potential, conductivity, mobility, density, surface tension, viscosity, Raman and FT-IR analyses were performed. Additionally, the UV-irradiated samples were analyzed in terms of their biological activities (antioxidant and antimicrobial effects).

### 3.7. Density, Surface Tension, and Viscosity Analyses

The density and surface tension of pure the *R. canina* seed oil, empty and oil-loaded liposomes (non-treated and UV-irradiated) were determined using silicon crystal as the immersion body and a Wilhelmy plate, respectively, in Force Tensiometer K20 (Kruss, Hamburg, Germany). Each sample (20 mL) was examined three times at 25 °C.

The viscosity of the pure *R. canina* seed oil, the empty, and the oil-loaded liposomes (non-treated and UV-irradiated) were examined using a Rotavisc lo-vi device equipment with a VOL-C-RTD chamber, VOLS-1 adapter, and spindle (IKA, Staufen, Germany). Each sample (6.7 mL) was examined three times at 25 °C.

The measurements of the density, surface tension, and viscosity were performed on the 1st and 60th days.

### 3.8. Raman and FT-IR Analyses

The micro-Raman spectra of the pure Phospholipon, the *R. canina* seed oil, the liquid and lyophilized plain and oil-loaded liposomes (non-treated and UV-irradiated samples) were collected in a backscattering configuration using a Jobin-Yvon T64000 triple spectrometer equipped with a nitrogen-cooled charge-coupled device detector (HORIBA Scientific, Kyoto, Japan). The spectral resolution was 2 cm^−1^ and the accuracy for all measured wavenumbers was ±3 cm^−1^. The spectra have been excited by a 514.5 nm line of Ar+/Kr+ ion laser with an output power of less than 2 mW to avoid the heating effects and/or sample thermal degradation.

The FT-IR spectra of the pure Phospholipon, *R. canina* seed oil, and lyophilized empty and oil-loaded liposomes (non-treated and UV-irradiated samples) were recorded in the transmission mode between 400 and 4000 cm^−1^, using a Nicolet iS10 spectrometer (Thermo Scientific, Gothenburg, Sweden).

### 3.9. Antioxidant Assays

#### 3.9.1. ABTS Assay

The ABTS method is based on the reduction of the ABTS•+ radical in an ethanol solution by antioxidants [70]. A mixture of the ABTS solution and 88 µL of a potassium persulfate solution was left to react for 24 h at 4 °C. Subsequently, 2 mL of the ABTS•+ solution was added to 20 µL of the liposomal samples (non-treated and UV-irradiated empty and oil-loaded liposomes). The free radical scavenging activity of the liposomes was evaluated by measuring the absorbance at 734 nm, after 6 min of incubation in the dark, compared to the blank and calculated as ∆A = A_c_-A_x_ (A_c_—the absorbance of ABTS•+ solution and water; A_x_—the absorbance of ABTS•+ solution and liposomes). All analyses were carried out in triplicate and the radical scavenging capacity was expressed as mmol Trolox equivalents per milliliter of the liposomal sample (µmol/mL), using the calibration curve of Trolox (0.2–1 mM).

#### 3.9.2. DPPH Assay

The antioxidant activity of the non-treated and UV-irradiated empty and *R. canina* oil-loaded liposomes, in terms of hydrogen donating or radical scavenging ability using the stable DPPH radical, was examined as well [70]. All measurements were performed as follows: the amount of 200 µL of five different concentrations of liposomes was added to 2.8 mL of the DPPH radical solution and the absorbance readings were taken after 20 min against the blank at 517 nm. The antioxidant capacity was calculated using the following equation: radical scavenging capacity (%) = (A_c_ − A_x_)*100/A_c_ (A_c_—the absorbance of the DPPH solution and water; A_x_—the absorbance of the DPPH solution and different liposomal concentrations). The results, obtained from triplicate analyses, were expressed as the concentration of the liposomes required to scavenge 50% of the free radicals, IC_50_ (mg/mL).

All absorbance readings were performed on a UV spectrophotometer, UV-1800 (Shimadzu, Kyoto, Japan).

### 3.10. Antimicrobial Analysis

The antimicrobial potential of the *R. canina* seed oil and the oil-loaded liposomes was investigated using three Gram-negative bacteria (*P. aeruginosa*, *Klebsiella* spp., and *Proteus* spp.), one Gram-positive bacteria (*S. aureus*), and one fungus (*C. albicans*), all isolated from wound swabs in the microbiology laboratory of the Institute for the Application of Nuclear Energy, INEP, Belgrade, Serbia. The antimicrobial activity was examined using the disk diffusion method [71]. The cultivation of the microorganisms was performed aerobically for 24 h on blood agar (for bacteria) and 48 h on Sabouraud agar (for fungus). The microorganism was incubated at 37 °C. The inoculum was prepared to an optical density of 0.5 McFarland (1.5 × 10^8^ CFU/mL) and the inoculum was spread on Petri dishes with Mueller Hinton agar (for bacteria) or Sabouraud agar (for fungus). Subsequently, 75 μL of the samples were transferred to the wells made in agar and incubated aerobically for 24 h at 37 °C. The criterion for detecting the inhibitory activity was the inhibition zone diameter.

### 3.11. Statistical Analysis

In the present study, the statistical analysis was performed using the analysis of variance (one-way ANOVA) followed by Duncan’s *post hoc* test, using the statistical software, STATISTICA 7.0. The differences were considered statistically significant at *p* < 0.05, n = 3.

## 4. Conclusions

Rosehip seed oil, as a valuable source of natural bioactive compounds, was encapsulated in phospholipid liposomes prepared using the proliposome technique. The vesicle sizes of the oil-loaded liposomes did not change significantly during the 60 days of storage, while a slight increase in the PDI values was noticed. The zeta potential and mobility varied in all liposomal populations, but the trend depended on the absence or the presence of rosehip oil and the UV irradiation. The conductivity of all liposomes significantly decreased during the 60-day stability study. The values of the density and surface tension of the oil-loaded liposomes did not change after 60 days, while the viscosity significantly decreased. Due to the possible synergistic beneficial effects of phospholipids and *R. canina* seed oil on human skin, the liposomal particles developed in the present study can find application in different cosmetic or pharmaceutical products. Since rosehip oil has shown a beneficial effect on the skin regeneration in traditional medicine, while the antimicrobial and significant antioxidant potential of the oil-loaded liposomes did not show in the present research, future studies should focus on the in vitro effects of the *R. canina* oil preparations on the migration and proliferation of skin cells in a wound scratch healing model.

## Figures and Tables

**Figure 1 molecules-28-00276-f001:**
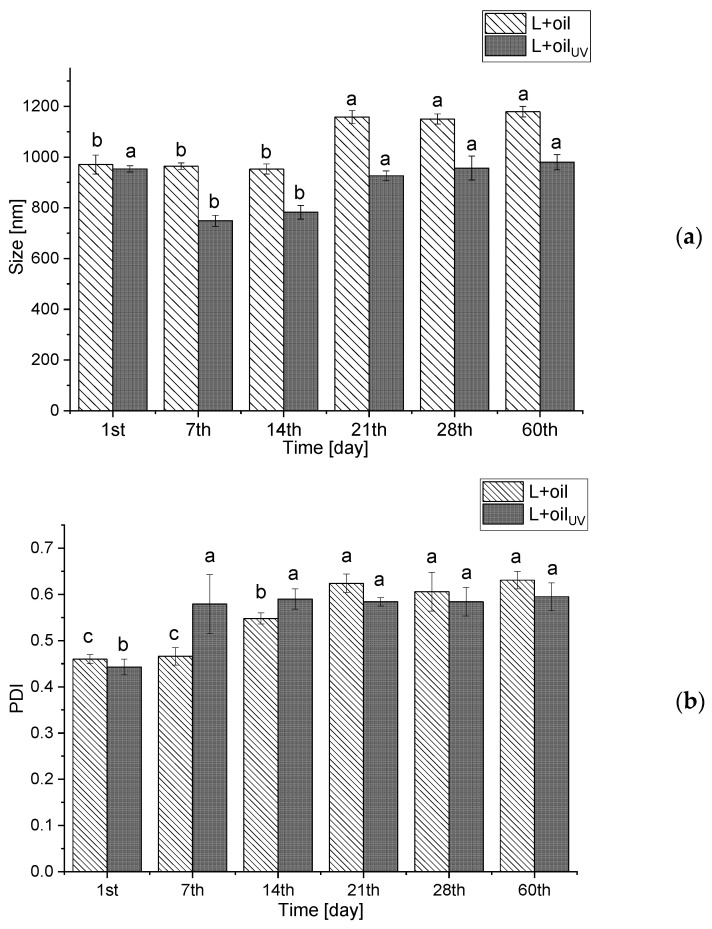
Liposome size (**a**), polydispersity index (**b**), zeta potential (**c**), mobility (**d**), and conductivity (**e**) of the non-treated and UV-irradiated *Rosa canina* seed oil-loaded liposomes during 60 days storage at 4 °C; values with different letters (**a**–**d**) in each row showed statistically significant differences (*p* < 0.05; n = 3; analysis of variance, Duncan’s *post-hoc* test); L, liposomes.

**Figure 2 molecules-28-00276-f002:**
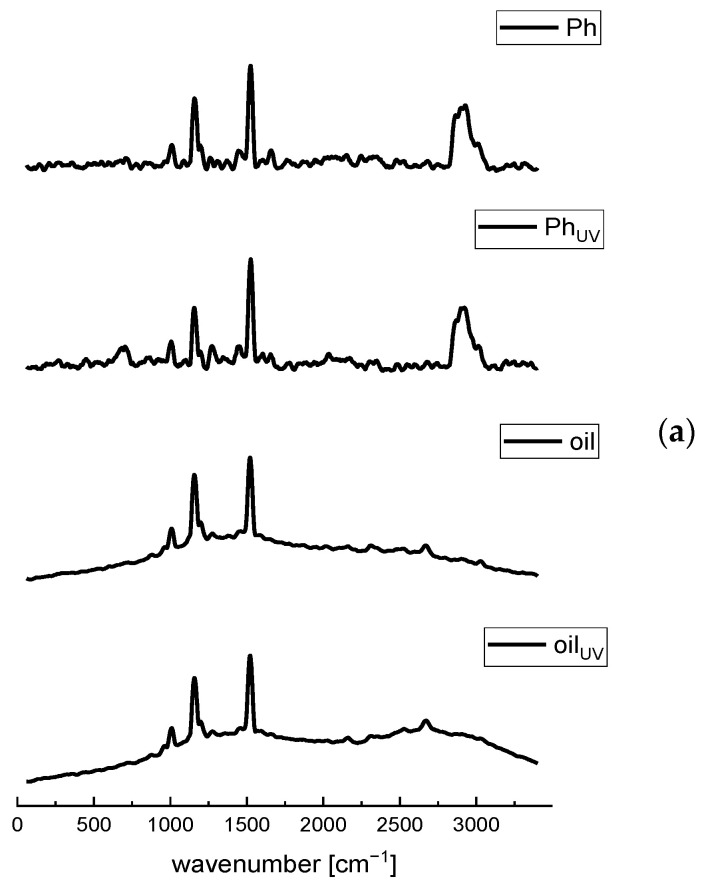
Raman spectra of the non-treated and UV-irradiated (**a**) Ph (a commercial phospholipids mixture) and the *Rosa canina* seed oil, and (**b**) the oil-loaded liposomes (liquid and lyophilized) in the spectral range from 70 to 3400 cm^−1^; L, liposomes.

**Figure 3 molecules-28-00276-f003:**
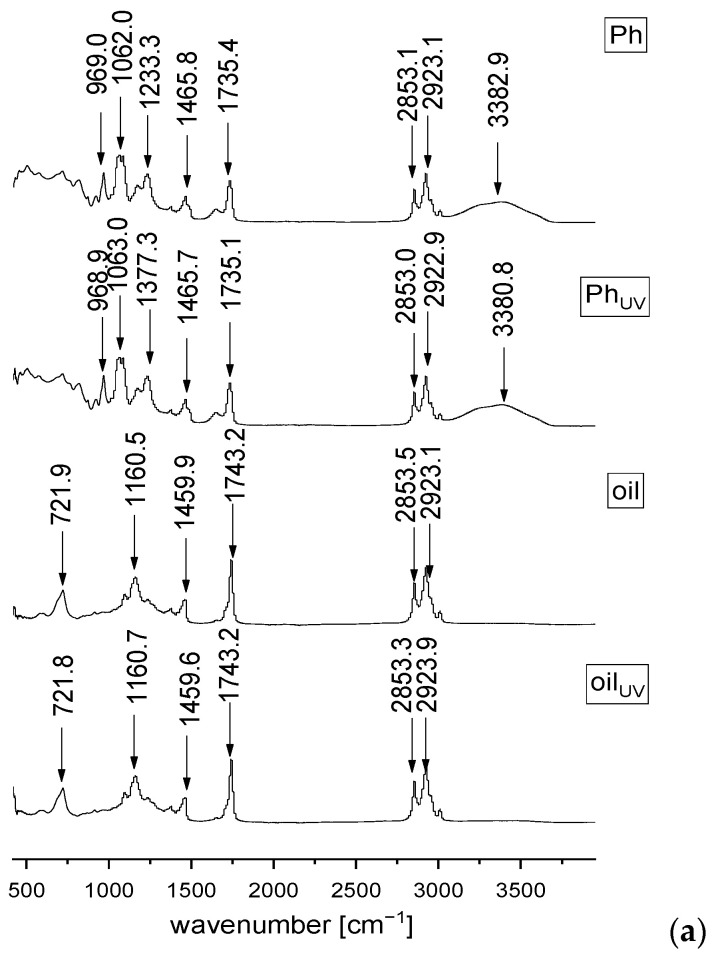
FT−IR spectra of the non-treated and UV-irradiated (**a**) Ph (a commercial phospholipids mixture) and *Rosa canina* seed oil, and (**b**) the plain and oil-loaded liposomes in the spectral range from 450 to 4000 cm^−1^; L, liposomes.

**Figure 4 molecules-28-00276-f004:**
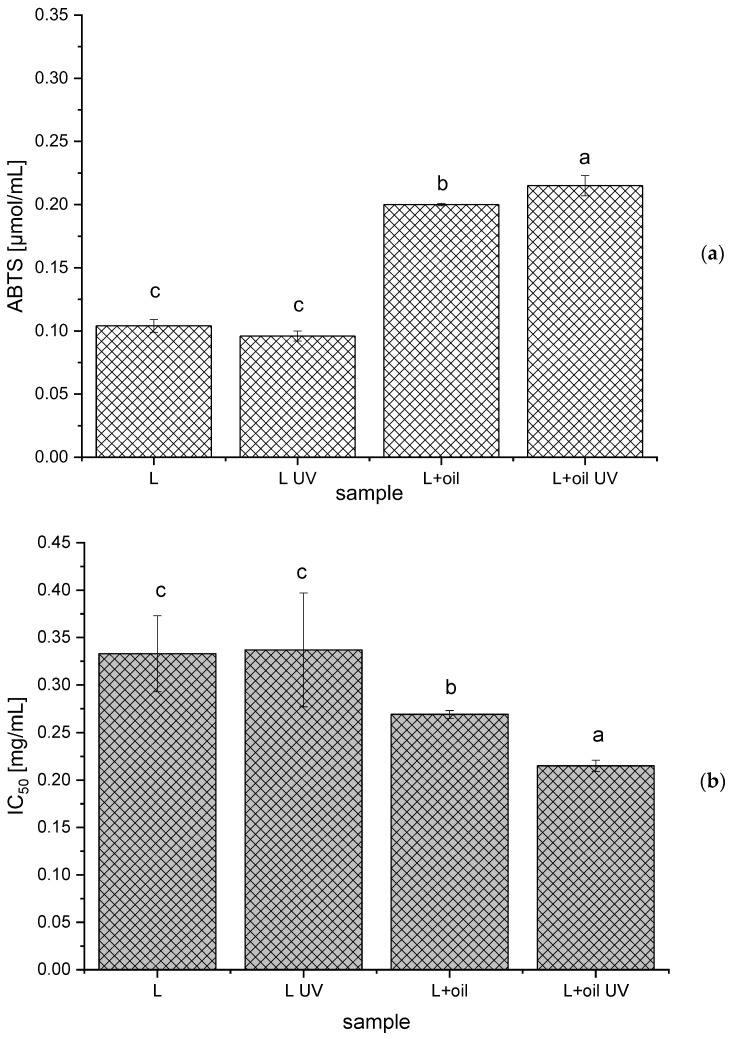
Antioxidant capacity of the empty and *Rosa canina* seed oil-loaded liposomes (non-treated and UV-irradiated) obtained in the ABTS (**a**) and DPPH (**b**) assays; IC_50_, the concentration of the liposomes required to scavenge 50% of the DPPH radicals; values with different letters (**a**–**c**) showed statistically significant differences (*p* < 0.05; n = 3; analysis of variance, Duncan’s *post-hoc* test).

**Table 1 molecules-28-00276-t001:** Particle size, polydispersity index (PDI), zeta potential (*ζ*), conductivity (*G*), and mobility (*µ*) of the empty and *Rosa canina* seed oil-loaded liposomes measured immediately after the preparation and after the UV irradiation.

Sample	Size [nm]	PDI	*ζ* [mV]	*G* [mS/cm]	*µ* [µmcm/Vs]
L	2145.7 ± 43.7 ^a^*	0.467 ± 0.012 ^a^	−22.7 ± 0.5 ^b^	0.020 ± 0.002 ^c^	−1.79 ± 0.03 ^b^
L_UV_	2127.3 ± 29.4 ^a^	0.472 ± 0.023 ^a^	−17.6 ± 0.3 ^c^	0.021 ± 0.002 ^c^	−1.22 ± 0.02 ^c^
L + oil	970.4 ± 37.4 ^b^	0.460 ± 0.010 ^a^	−32.9 ± 0.8 ^a^	0.068 ± 0.002 ^a^	−2.58 ± 0.06 ^a^
L + oil_UV_	953.4 ± 12.2 ^b^	0.443 ± 0.017 ^a^	−21.9 ± 0.6 ^b^	0.032 ± 0.001 ^b^	−1.72 ± 0.04 ^b^

* Values with different letters (a–c) in each row showed statistically significant differences (*p* < 0.05; n = 3; analysis of variance, Duncan’s *post-hoc* test); L, liposomes.

**Table 2 molecules-28-00276-t002:** Density (*ρ*), surface tension (*γ*), and viscosity (*η*) of *Rosa canina* seed oil, empty, and oil-loaded liposomes (non-treated and UV-irradiated).

Sample	*ρ* (g/cm^3^)	*γ* (mN/m)	*η* (mPa·s)
oil	0.914 ± 0.001 ^d^*	30.0 ± 0.8 ^a^	50.9 ± 0.3 ^a^
oil_UV_	0.915 ± 0.002 ^d^	28.0 ± 0.9 ^b^	45.7 ± 0.1 ^b^
L	0.995 ± 0.003 ^a^	19.7 ± 0.7 ^e^	13.9 ± 0.2 ^d^
L_UV_	0.967 ± 0.003 ^b^	21.4 ± 0.5 ^d^	20.7 ± 0.4 ^c^
L + oil	0.974 ± 0.004 ^b^	20.3 ± 0.8 ^de^	13.5 ± 0.2 ^d^
L + oil_UV_	0.959 ± 0.001 ^c^	23.1 ± 0.2 ^c^	20.2 ± 0.2 ^c^

* Values with different letters (a–e) in each row showing the statistically significant differences (*p* < 0.05; n = 3; analysis of variance, Duncan’s *post-hoc* test); L, liposomes.

## Data Availability

The datasets generated during and/or analyzed during the current study are available from the corresponding author upon reasonable request.

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
