# Peer review of "Liposomal Bilayer as a Carrier of Rosa canina L. Seed Oil: Physicochemical Characterization, Stability, and Biological Potential"

_molecules, 2022, doi:10.3390/molecules28010276_

Round 1

Reviewer 1 Report

Title: Liposomal bilayer as a carrier of Rosa canina L. seed oil: physicochemical characterization, stability, and biological potential

Journal: Molecules

 This study prepared liposomes delivered functional oil and evaluated its physicochemical characterization, stability, and biological potential. Sufficiency data were collected and analysis. However, many places need to be improved before it is accepted. Detailed comments are shown below.

1.     Line 18: change ‘are’ to ‘were’.

2.     Line 22-23: are you sure the oil distributed both inside of liposome and liposomal surface? The core of liposomes carries hydrophilic compounds.

3.     Line 26: add ‘of storage’ after ‘days’.

4.     Line 28: change ‘should’ to ‘could’.

5.     There are too much keywords. The author should refine them.

6.     The first paragraph of introduction is too long to catch the main content. It should be separated to at least two paragraphs. Besides, based on current information of this paragraph, it is hard to get the key information. Too much sentences talked about the function of Rosehips oil, and the cited references also provided too much. However, why the oil needs to be encapsulated is not clear stated.

7.     Line 95-108: Raman and Fourier transform infrared (FT-IR) spectroscopy are introduced too much. They are the based tools for the evaluation of delivery system. There is no need to write so much. On the contrary, I didn’t see enough literature reviews on the liposomal delivery for functional oil compounds.

8.     The last paragraph should provide the information on the aim of this study.

9.     Figure 1: the first letter of the title in each coordinate should be the uppercase. I suggested to remove the data in the figure. Besides, why the conductivity was measured during storage but not the antioxidative ability changes or other factors?

10.  Table 1: what is the conductivity of liposomes uses for?

11.  Why there is no results on encapsulated efficiency?

12.  Figure 3: the key wavelength should be indicated in the figure.

13.  Where is the results of antimicrobial potential of liposomes?

Author Response

Dear Reviewer,  The authors would like to thank you for the time and efforts that improved the manuscript molecules-2067277 entitled ''Liposomal bilayer as a carrier of Rosa canina L. seed oil: physicochemical characterization, stability, and biological potential''. We have thoroughly revised our manuscript taking into account all your recommendations. We hope that the manuscript has been improved and is acceptable for publication. Here are our answers to specific points: 

Reviewer 1

This study prepared liposomes delivered functional oil and evaluated its physicochemical characterization, stability, and biological potential. Sufficiency data were collected and analysis. However, many places need to be improved before it is accepted. Detailed comments are shown below.

Point 1: Line 18: change ‘are’ to ‘were’.

Response 1: As suggested, the word ‘are’ was changed to ‘were’.

Point 2: Line 22-23: are you sure the oil distributed both inside of liposome and liposomal surface? The core of liposomes carries hydrophilic compounds.

Response 2: Regarding the Reviewer's suggestion, the sentence was reformulated, and instead of ''inside and on the liposomal surface'', we wrote ''inside the phospholipid bilayer and combined with the membrane interface of the bilayer'', as a more frequently used phrase in literature. Additionally, the literature data and explanations which confirm the claim are given in the text (Section 2.5. FT-IR spectra), as well as the reference in the reference list (Song, F., Tian S., Yang, G., Sun X. Effect of phospholipid/flaxseed oil ratio on characteristics, structure change, and storage stability of liposomes. LWT 2022, 157, 113040. https://doi.org/10.1016/j.lwt.2021.113040).

Point 3: Line 26: add ‘of storage’ after ‘days’.

Response 3: As suggested, ‘of storage’ were added after ‘days’.

Point 4: Line 28: change ‘should’ to ‘could’.

Response 4: As suggested, it was changed.

Point 5: There are too much keywords. The author should refine them.

Response 5: According to the Instructions for Authors of the journal Molecules, three to ten pertinent keywords need to be added after the abstract, thus we had 9 keywords. However, after the Reviewer's suggestion, we reduced the number of keywords (5 keywords).

Point 6: The first paragraph of introduction is too long to catch the main content. It should be separated to at least two paragraphs. Besides, based on current information of this paragraph, it is hard to get the key information. Too much sentences talked about the function of Rosehips oil, and the cited references also provided too much. However, why the oil needs to be encapsulated is not clear stated.

Response 6: According to the Reviewer's suggestions, the first paragraph of the Introduction was shortened and separated into three paragraphs. Additionally, the reasons for the encapsulation of Rosa canina seed oil and the advantages of the encapsulated products are separated and presented as the third paragraph of the Introduction (oil bioactive compounds' sensitivity to oxygen, light, UV irradiation, enzymes, and pH value variations, their low bioavailability, literature data that claims their sensitivity). Also, one more sentence was added. We hope that after the mentioned changes, it is clearly stated why the oil was encapsulated.

Point 7: Line 95-108: Raman and Fourier transform infrared (FT-IR) spectroscopy are introduced too much. They are the based tools for the evaluation of delivery system. There is no need to write so much. On the contrary, I didn’t see enough literature reviews on the liposomal delivery for functional oil compounds.

Response 7: According to the Reviewer's suggestions, the section on Raman and Fourier transform infrared (FT-IR) spectroscopy was significantly shortened and now it contains only one sentence with appropriate references. The information related to the studies dealing with liposomal encapsulation of fatty oils was added as well (as one sentence in the Introduction and three references in the reference list).

Point 8: The last paragraph should provide the information on the aim of this study.

Response 8: According to the Reviewer's suggestion, the information on the aim of this study was added in the last paragraph.

Point 9: Figure 1: the first letter of the title in each coordinate should be the uppercase. I suggested to remove the data in the figure. Besides, why the conductivity was measured during storage but not the antioxidative ability changes or other factors?

Response 9: According to the Reviewer's suggestions, the first letter of the title in each coordinate was uppercase and the data was removed from the figure. Therefore, now Figure 1 contains 5 graphics (a-e). The reasons for monitoring conductivity are added in Sections 2.1. Size, PDI, zeta potential, conductivity, and mobility of non-treated and UV-irradiated liposomes and 2.2. The storage stability of liposomes. Apart from the conductivity, particle size, PDI, zeta potential, and mobility were also monitored during 60 days stability study, because according to the literature, the mentioned parameters are widely used as a measure of system instability (Lidgate et al., 1993. International Journal of Pharmaceutics 96, https://doi.org/10.1016/0378-5173(93)90211-W; Lopes de Azambuja et al., 2015. Chemistry and Physics of Lipids 193, https://doi.org/10.1016/j.chemphyslip.2015.10.001; Nakhaei et al. 2021. Front Bioeng Biotechnol. 9, https://doi.org/10.3389/fbioe.2021.705886). Additionally, antioxidant ability was not monitored during 60 days of storage, because the antioxidant capacity of oil-loaded liposomes was significantly low already on the 1st day. Namely, since rosehip oil has shown a beneficial effect on skin regeneration in traditional medicine, while the antimicrobial and significant antioxidant potential of the oil-loaded liposomes have not been demonstrated in the present research, future studies should focus on the in vitro effects of R. canina oil preparations on the migration and proliferation of skin cells in wound scratch healing model. The density, surface tension, and viscosity measurements of the non-treated and UV-irradiated oil-loaded liposomes were performed at 1st and 60th days, because of the availability of the devices, which belong to another department, thus measurements could not be performed as often as in the case of photon correlation spectroscopy (1st, 7th, 14th, 21st, 28th, and 60th day). The values of the density, surface tension, and viscosity measurements on the 60th day were added to the manuscript (Section 2.3. Density, surface tension, and viscosity of the liposomes).

Point 10: Table 1: what is the conductivity of liposomes uses for?

Response 10: The reasons for monitoring conductivity were added in Sections 2.1. Size, PDI, zeta potential, conductivity, and mobility of non-treated and UV-irradiated liposomes and 2.2. The storage stability of liposomes.

Point 11: Why there is no results on encapsulated efficiency?

Response 11: Since we have investigated the activity of the whole liposomal formulation, not only the part without the supernatant (with non-encapsulated compounds), as well as we did not deal with the quantification of the individual compounds of the oil, thus we did not determine the encapsulation efficiency of the oil compounds, as in the case of several studies which dealt with the encapsulation of oil into liposomes (Huang et al., 2020. LWT 117, https://doi.org/10.1016/j.lwt.2019.108615; Li et al., 2020. Food Science and Human Wellness 9, https://doi.org/10.1016/j.fshw.2020.01.002; Song et al., 2023. Food Chemistry 404, https://doi.org/10.1016/j.foodchem.2022.134547). Additionally, in the preliminary study, we investigated the maximum loading capacity of the liposomes, i.e. the amount of oil that can be encapsulated into liposomes. Namely, in the preliminary screening, the amount of oil in the liposomal preparation was varied (0.5-1.5 mL) and after centrifugation, in the case of samples prepared using 0.5-1.2 mL of oil, there were no oil drops on the surface of the supernatant (Figure A). Therefore, 1.2 mL of the oil was selected for the preparation of future liposomes that were further examined. The authors used the samples after the centrifugation and without the supernatant in the FI-IR and Raman spectroscopy analyses.

Figure A Rosa canina seed oil-loaded liposomes (with 1.2 mL of the oil) after centrifugation.

Point 12: Figure 3: the key wavelength should be indicated in the figure.

Response 12: As suggested, the key wavelengths were indicated in Figure 3.

Point 13: Where is the results of antimicrobial potential of liposomes?

Response 13: Since neither pure oil nor oil-loaded liposomes showed antimicrobial potential, the authors decided not to show the results, i.e. the pictures made during the disk diffusion method in the version submitted to the journal. However, in this response you can find the picture represent as Figure B:

Figure B Antimicrobial activity of pure Rosa canina seed oil and oil-loaded liposomes (non-treated and UV-irradiated).

The authors also carefully considered and answered all issues mentioned in the Review Report Form, therefore improving the English language and style, Introduction, and references relevant to the research, as well as the methods' description and presentation of the results.

Reviewer 2 Report

Comments to authors;

The authors described the manuscript entitled “Liposomal bilayer as a carrier of Rosa canina L. seed oil: physicochemical characterization, stability, and biological potential. Below you can find detailed description. The study provides with the standard characterization of delivery systems. I find this work valuable and interesting. Nevertheless, I suggest authors to emphasize the scientific importance of their study. It would be beneficial to the manuscript if authors improve the part concerning the purpose of their study expressing both the novelty and scientific weight of the study.

Author Response

Dear Reviewer,  The authors would like to thank you for the time and efforts that improved the manuscript molecules-2067277 entitled ''Liposomal bilayer as a carrier of Rosa canina L. seed oil: physicochemical characterization, stability, and biological potential''. We have thoroughly revised our manuscript taking into account all your recommendations. We hope that the manuscript has been improved and is acceptable for publication. Here are our answers to specific points: 

Reviewer 2

The authors described the manuscript entitled “Liposomal bilayer as a carrier of Rosa canina L. seed oil: physicochemical characterization, stability, and biological potential. Below you can find detailed description. The study provides with the standard characterization of delivery systems. I find this work valuable and interesting.

Point 1: Nevertheless, I suggest authors to emphasize the scientific importance of their study. It would be beneficial to the manuscript if authors improve the part concerning the purpose of their study expressing both the novelty and scientific weight of the study.

Response 1: According to the Reviewer's suggestions, the authors improved the whole Introduction with the aim to emphasize the scientific importance of the study and present the novelty and scientific weight of the study. Additionally, the authors also improved the presentation of the results.

Reviewer 3 Report

This article has described the synthesis and characterization of liposomes loaded with natural bioactive components from R. canina oil. The aim of the R. canina oil incorporation into the liposomal composition was to protect its bioactive components from damage caused by UV-irradiation. The oil-loaded liposomes developed have shown stability against UV-irradiation and antioxidant activity. The liposomes have been thoroughly characterized by a large number of experimental techniques, and the results obtained have been discussed comprehensively. As for comments and suggestions, several points are listed below.

 Comments and Suggestions for Authors

1. Regarding the R. canina oil loading: how the oil-loaded liposomes were separated from the oil unbound? What was the maximum loading capacity of the liposomes developed?

2. What was the antimicrobial/antioxidant activity of the pristine R. canina oil under the same conditions, especially during the disk diffusion experiments? Such comparison seems to be very important to evaluate influence of the liposome surroundings on the bioactivity of the R. canina oil components.

3. Authors have observed a 25%-decrease in the conductivity of all liposomes after a week of storage at 4°C, however, the others liposomal characteristics have remained nearly the same during 2-months of storage at 4°C. What was the reason behind these different experimental tendencies? How does the storage affect the antioxidant properties of the oil-loaded liposomes?

On the whole, the manuscript submitted represents a high-level experimental study of a liposome nanocarrier and contains new valuable results which can be useful in pharmaceutical and cosmetic field. Thus, I recommend this manuscript to be published in “Molecules”.

Author Response

Dear Reviewer,  The authors would like to thank you for the time and efforts that improved the manuscript molecules-2067277 entitled ''Liposomal bilayer as a carrier of Rosa canina L. seed oil: physicochemical characterization, stability, and biological potential''. We have thoroughly revised our manuscript taking into account all your recommendations. We hope that the manuscript has been improved and is acceptable for publication. Here are our answers to specific points:

Reviewer 3

This article has described the synthesis and characterization of liposomes loaded with natural bioactive components from R. canina oil. The aim of the R. canina oil incorporation into the liposomal composition was to protect its bioactive components from damage caused by UV-irradiation. The oil-loaded liposomes developed have shown stability against UV-irradiation and antioxidant activity. The liposomes have been thoroughly characterized by a large number of experimental techniques, and the results obtained have been discussed comprehensively. As for comments and suggestions, several points are listed below.

Comments and Suggestions for Authors

Point 1: Regarding the R. canina oil loading: how the oil-loaded liposomes were separated from the oil unbound? What was the maximum loading capacity of the liposomes developed?

Response 1: In the preliminary study, we investigated the maximum loading capacity of the liposomes, i.e. the amount of oil that can be encapsulated into liposomes. Namely, in the preliminary screening, the amount of oil in the liposomal preparation was varied (0.5-1.5 mL) and after centrifugation, in the case of samples prepared using 0.5-1.2 mL of oil, there were no oil drops on the surface of the supernatant (Figure A). Therefore, 1.2 mL of the oil was selected for the preparation of future liposomes that were further examined. In the FI-IR and Raman spectroscopy analyses, we used the samples after the centrifugation (17,500 rpm and 4°C for 45 min in Thermo Scientific Sorval WX Ultra series ultracentrifuge, ThermoScientific, USA) and without the supernatant. All mentioned information were also added to the manuscript (Section 3.3. Preparation of liposomal particles and lyophilization).

Figure A Rosa canina seed oil-loaded liposomes (with 1.2 mL of the oil) after centrifugation.

Point 2: What was the antimicrobial/antioxidant activity of the pristine R. canina oil under the same conditions, especially during the disk diffusion experiments? Such comparison seems to be very important to evaluate influence of the liposome surroundings on the bioactivity of the R. canina oil components.

Response 2: The antioxidant activity of pure R. canina oil under the same conditions was added in Section 2.6. The antioxidant potential of the liposomes, while the antimicrobial activity of pure R. canina oil under the same conditions in the disk diffusion experiments has already been mentioned in the manuscript (Abstract and 3.7. Antimicrobial potential of the liposomes): ''However, neither R. canina seed oil nor oil-loaded liposomes showed antimicrobial activity against all investigated microorganisms.'' In this document, the results of the antimicrobial potential of pure oil and oil-loaded liposomes are represented as Figure B:

Figure B Antimicrobial activity of pure Rosa canina seed oil and oil-loaded liposomes (non-treated and UV-irradiated).

Since neither pure oil nor oil-loaded liposomes showed antimicrobial potential, the authors decided not to show the results, i.e. the pictures made during the disk diffusion method in the version submitted to the journal.

Point 3: Authors have observed a 25%-decrease in the conductivity of all liposomes after a week of storage at 4°C, however, the others liposomal characteristics have remained nearly the same during 2-months of storage at 4°C. What was the reason behind these different experimental tendencies? How does the storage affect the antioxidant properties of the oil-loaded liposomes?

Response 3: Due to the Reviewer's significant observation and suggestion, the additional paragraph was added to Section 2.2. The storage stability of liposomes, with the aim to improve the manuscript and better explain the obtained results and phenomenon: ''According to the Lidgate et al. [41] study, the measurement of conductivity of the stored liposomal vesicles can provide important information related to their size and integrity, i.e. potential leaking of the encapsulated substances. Namely, changes in conductivity can indicate whether liposomes have fused, i.e. the decrease in conductivity, or leaked i.e. increase in conductivity. ... As can be noticed, the conductivity of oil-loaded liposomes decreased on the 14th day, after that the values did not change. A similar trend was observed in the PDI values of the mentioned liposomes (the increase in PDI on the 14th day, after that the values were the same). Therefore, the decrease in conductivity can be related to the fusion of some parts of the liposomal population, which influenced the size distribution, but not particle size represented as mean value (Figures 1a-b).''

The antioxidant activity of the oil-loaded liposomes did not measure during the storage, because their antioxidant capacity was significantly low already on the 1st day. According to the literature, R. canina seed oil is used in traditional medicine as a skin regeneration agent, but probably due to its effects on cell migration and proliferation, which will be examined in future experiments, since the present research did not show significant antioxidant activity.

Point 4: On the whole, the manuscript submitted represents a high-level experimental study of a liposome nanocarrier and contains new valuable results which can be useful in pharmaceutical and cosmetic field. Thus, I recommend this manuscript to be published in “Molecules”.

Response 4: The authors also improved the conclusions (the issue mentioned in the Review Report Form) by accepting all Reviewer's suggestions.

Round 2

Reviewer 1 Report

The manuscript has been improved after the revision. It can be accepted at current stage.